METHODS

# StellarPath: Hierarchical-vertical multi-omics classifier synergizes stable markers and interpretable similarity networks for patient profiling

**Luca Giudice**[ID]*, **Ahmed Mohamed**[ID], **Tarja Malm**[ID]

A.I. Virtanen Institute for Molecular Sciences, University of Eastern Finland, Kuopio, Finland

* luca.giudice@uef.fi

**Data Availability Statement:** StellarPath is implemented in R with the graph convolutional network made in Python interacting through the Reticulate library. We provide a GitHub repository

## Abstract

The Patient Similarity Network paradigm implies modeling the similarity between patients based on specific data. The similarity can summarize patients' relationships from high-dimensional data, such as biological omics. The end PSN can undergo un/supervised learning tasks while being strongly interpretable, tailored for precision medicine, and ready to be analyzed with graph-theory methods. However, these benefits are not guaranteed and depend on the granularity of the summarized data, the clarity of the similarity measure, the complexity of the network's topology, and the implemented methods for analysis. To date, no patient classifier fully leverages the paradigm's inherent benefits. PSNs remain complex, unexploited, and meaningless. We present StellarPath, a hierarchical-vertical patient classifier that leverages pathway analysis and patient similarity concepts to find meaningful features for both classes and individuals. StellarPath processes omics data, hierarchically integrates them into pathways, and uses a novel similarity to measure how patients' pathway activity is alike. It selects biologically relevant molecules, pathways, and networks, considering molecule stability and topology. A graph convolutional neural network then predicts unknown patients based on known cases. StellarPath excels in classification performances and computational resources across sixteen datasets. It demonstrates proficiency in inferring the class of new patients described in external independent studies, following its initial training and testing phases on a local dataset. It advances the PSN paradigm and provides new markers, insights, and tools for in-depth patient profiling.

## Author summary

A clinician's decision-making process for diagnosing an unknown patient involves selecting properties where patients with the same condition are alike but distinct from others, creating a mental database of known patients linked by their similarities, and evaluating the condition of the unknown patient based on its similarities with those in the database. However, constructing a network from biological omics data, which profiles patients through thousands of molecules, poses a significant challenge. The network's quality

with many vignettes: https://github.com/LucaGiudice/StellarPath We also provide a Zenodo repository that includes all the code and data used for the project: https://zenodo.org/record/8318554 All other relevant data are within the manuscript and its Supporting information files.

**Funding:** LG salary has been funded by Business Finland [6478/31/2019] and the Academy of Finland [339763]. TM was supported by Business Finland [6478/31/2019] and the Academy of Finland [339763]. AM received salary from the European Union's Horizon 2020 research and innovation programme under the Marie Skłodowska-Curie agreement [101034307] (NEURO-INNOVATION: Research and innovation for brain health throughout life). The funders had no role in study design, data collection and analysis, decision to publish, or preparation of the manuscript.

**Competing interests:** The authors have declared that no competing interests exist.

hinges on the choice of the molecules used to determine the patients' similarities, the significance of the similarity measure, and the network's structure interpretability. StellarPath is a patient classifier based on patient similarity networks. It manages data normalization, molecule selection, and combines molecule types based on their interactions. It determines key cellular functions distinguishing patient classes and models how much two patients are similar based on how their cells regulate each function. It retains networks that group similar patients while separating different ones, using this data to predict unknown patients' outcomes. Tested on seventeen datasets, StellarPath has proven efficient and insightful. It pushes the patient similarity network paradigm forward, offers reliable markers, and provides new tools and insights for patient analysis.

## Introduction

Patient Similarity Networks (PSN), where nodes represent patients and edges represent their similarities based on clinical or biological data, offer a new approach to modeling data for precision medicine [1]. These similarities can be used for tasks, such as training un/supervised machine learning algorithms, stratifying patient groups, identifying outliers, and revealing hidden subgroups.

One of the main advantages of this paradigm is that similarities can be employed to predict the class of unknown patients. In this supervised classification framework, given a set of known patients divided into classes (e.g., "lung cancer" and "healthy"), the goal is to identify key features and rules to classify new patients. For instance, a network based on cigarette smoking frequency might serve as a predictive feature, with the rule being that a patient is predicted to have lung cancer if its smoking habit similarly match those of known lung cancer patients.

The selection of the proper features is crucial for patient classification, as well as when PSNs are involved. An algorithm can find patterns and predict correctly with spurious or contradicting patients' features, which can lead to unreliable results. As a consequence, the selection of meaningful clinical and biological features as well as the classification itself is often undertaken manually by experts.

For instance, studies by Collisson et al. [2], Moffitt et al. [3], and Bailey et al. [4] on Pancreatic Ductal Adenocarcinoma (PDA) patients used significantly enriched pathways to manually classify patients and identify overlaps between their cohorts, underscoring the importance of biological processes in finding commonalities across different studies.

Pathways, such as biological processes or cellular functional sets, are often used by human experts and have proven to be invaluable for patient classification [5,6]. They are more robust to noise and interpretable than individual molecules, enhance the consistency of findings and natively support the integration of multiple omics. In fact, by their nature, pathways model multiple cellular elements and provide an intuitive summary of different molecules' roles [7,8].

In this direction, the PSN paradigm, leveraging the pathway space, could create an interpretable patient classifier. A predictive PSN could be based on how much the patients similarly regulate the Pulmonary Oxidative Stress. Thus, the lung cancer patients would be strongly similar because smoking induces the upregulation of stress signalling cellular pathways, while the healthy patients would be dissimilar because their cells would not be coordinated to regulate those pathways with the same intensity (i.e. gene expression, protein abundance, . . .).

Another main advantage of the paradigm is that a PSN summarizes two patients' vectors of information into one similarity value and is naturally represented as a graph. For these

reasons, a PSN can be more interpretable than the original dataset which contains the patient's profiles (e.g. gene expression matrix). However, the interpretability of a PSN is contingent upon two critical factors: the similarity measure and the network's topology. While the strength of similarity between two patients might be visually represented by the thickness of their connecting edge, the interpretability can be compromised if the underlying similarity measure is intricate or not transparent. In parallel, a network topology that blurs the distinctions between patient classes or becomes overly intricate can hinder clear interpretation. For instance, if a PSN fails to distinctly separate lung cancer patients from healthy ones due to an ambiguous topology, it becomes less insightful than a straightforward heatmap representation of the original dataset.

Consequently, the virtue and usefulness of a PSN-based patient classifier strongly depend on the quality of the features, the similarity measure, and the interpretability of the networks.

The recent review by Gliozzo et al. [9] lists classifiers based on the PSN paradigm. It highlights that, to date, the only published method for patient classification is netDx [10,11]. netDx does not perform feature selection, employs the Pearson correlation as similarity measure for patient profiles described with biological omics, and does not look for interpretable networks. It creates PSNs from datasets, merges them into one consensus network, and uses GeneMANIA for patient classification. According to Gliozzo et al., netDx uses a parallel integration flow and an early integration strategy. It implies that netDx does not model the inter-relationships among omics (e.g., the anti-correlation between miRNA and its target genes). Instead, netDx leans on GeneMANIA for the integration of multiple omics. GeneMANIA treats the omics as independent views without delving into their biological significance, combines them to generate one predictive consensus PSN, trains on the latter, and subsequently predicts unseen patients. netDx is designed to be versatile and can be configured by the user to generate PSNs from various types of data for the classification of the patients of interest [11,12].

The design of netDx is prevalent among patient classifiers in the literature that handle biological omics because guarantees versatility. The drawbacks such as hyper-parameters, user's required inputs, and computational resources are annotated by Fabris et al. [13]. Consequently, enrichment algorithms, which identify significant features based on predetermined rules without evaluating their predictive efficacy, remain the primary tools for biologists, clinicians, and bioinformaticians [14–16]. These tools provide qualitative biological evidence to discern patients' classes while remaining simple to apply, computationally efficient, and easy to understand [17,18].

Overall, creating a PSN-based classifier for patient prediction encompasses several challenges. It involves identifying meaningful biological features, and avoiding reliance on features that are predictive but biologically meaningless. Integrating diverse omics to reflect a biologist's expertise, steering clear of artificially feeding fused omics into a model. Ensuring that PSNs are both interpretable and reflective of significant biological patterns for methodological transparency. Providing PSNs that enable the retrieval of new meaningful insights for further analyses such as patient stratification, outlier detection and correlation with clinical data. Ultimately, overcoming existing classifiers' drawbacks, such as extensive user input, hyper-parameter tuning, and high computational demands.

We posit that there is a need for a patient classifier that fully harnesses the advantages of the PSN paradigm and matches enrichment methods. Such a classifier should select relevant features from multiple omics, integrate them in a manner that mirrors a biologist's expertise, employ a similarity measure with clear biological significance, identify predictive PSNs that are interpretable, and accurately classify patients. Furthermore, it should provide new insights about the patients that enable subsequent analyses like stratification.

We introduce StellarPath, a deep learning classifier that synergizes pathway analysis with patient similarity concepts in a hierarchical-vertical integration flow with a late integration strategy. StellarPath analyses biological omics with both conventional and novel approaches, models the inter-relationships among them into pathways, uses a novel patient similarity measure to build pathway-specific PSNs, keeps only networks with an interpretable and biologically meaningful topology, employs a graph convolutional network (GCN) to train how to recognize patients and predict their class. Finally, our method includes a new subgroup cohesive detection algorithm to reduce the complexity of the significant resulting PSNs to ease their visualization and interpretation.

StellarPath advances the work of netDx to bridge the gap between machine learning algorithms and precision medicine, focusing on the interpretability and the biological relevance of the features to maximize the classifier's practicality. At the same time, it moves forward the patient similarity network paradigm and introduces completely new concepts regarding the analysis of biological omics data.

## Methods

### Overview

StellarPath is a binary classifier, that processes high-throughput omics data describing patients in comparison, identifies features that distinguish the two classes, and trains Graph Convolutional Networks (GCNs) on such features to predict the class of unknown patients (Fig 1).

At the start, it divides the patients into three sets: training, validation, and testing. Training patients are known examples of the classes, are used to find relevant features (molecules, pathways, and PSNs) and to train the GCNs in classifying. The validation set is reserved to evaluate the efficacy that each GCN reached with the training. Testing patients are considered unknown, are introduced only when the training phase is finished, and are used to test the final efficacy of the GCNs.

StellarPath analyses each ome individually to determine the best differentially expressed and differentially stable molecules between the classes. It hierarchically combines the lists of significantly deregulated molecules and finds the over-represented pathways.

StellarPath builds pathway-specific PSNs. For each enriched pathway, StellarPath uses a novel similarity to assess how much each pair of training patients is similar. The similarity evaluates how much the patient's values in the significant molecules of the pathway are close and similarly deregulated. The similarities are then used to build one PSN that represents the relationships between patients in one specific pathway. Next, a PSN is kept and considered a signature of a class only if showing a specific topology. Patients within one class must exhibit strong similarities to each other, while the patients in the opposite class must not be similar. Additionally, there must be clear dissimilarity between members of the two classes (Fig 1A).

At this stage, StellarPath examines the patients using the pathway-specific class-signature PSNs. It measures the centrality (i.e. importance) of each patient based on its similarities in every PSN. This allows the user to detect which pathways are most relevant to individual patients within a class (Fig 1D).

Finally, StellarPath uses the PSNs to train GCNs. Once trained, the validation patients are integrated into the PSNs and their classes are predicted by each GCN based on their similarity to the training patients. After the training, the testing patients are included and each GCN predicts their classes. The final class prediction for the testing patients is determined by majority voting (i.e. Ensemble).

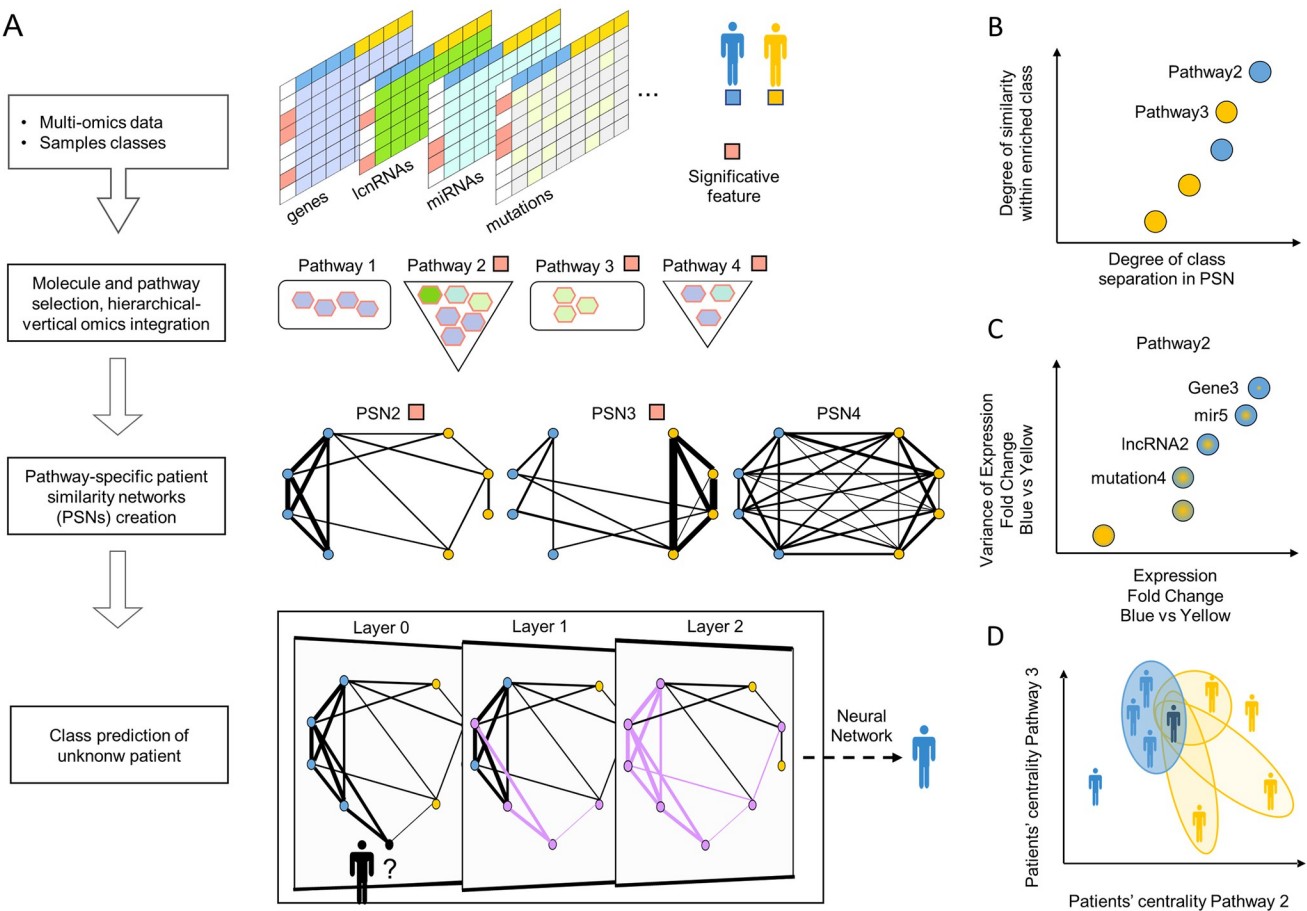

**Fig 1. Overall scheme of StellarPath.** (A) Workflow of StellarPath. The method classifies two patients' classes and works with both dense and sparse (e.g., somatic mutation) omics. It finds the significantly deregulated molecules and their enriched pathways. It determines how much each pair of patients is similar by comparing the values of the molecules belonging to a specific pathway. It uses pathway-specific similarities to build a network. Only the PSNs, which show that one class is cohesive while the opposite one is not, are kept. StellarPath receives an unknown patient, adds it to a significant PSN, and predicts its class with a graph convolutional network. StellarPath provides multiple output data: (B) significant pathway-specific PSNs provide how much the classes in comparison are separated (x-axis), how strong the cohesion (y-axis) within the cohesive class (color) (C) a PSN represents the pathway's similarities due to differentially expressed and differentially stable molecules (D) both known and unknown patients are stratified and quality checked based on how much are similar in the pathways.

## Data preparation

StellarPath requires one or more high-throughput datasets (i.e. biological omics describing the same samples). These datasets must be in a continuous numerical matrix format, where each row represents a different molecule and each column corresponds to a patient's sample. The patient samples must be divided into two distinct classes to compare, such as cases and controls in a differential expression analysis setting. Thereafter, StellarPath automatically incorporates the list of pathways describing functional associations and interactions between the molecules in the omics from the MSigDB [19], GO [20], Kegg [21], miRWalk [22], RNAInter [23] and OmniPath [24] databases; this means that it can model different types of pathways like cellular functions (e.g. Hallmark Glycolysis), signaling cascades (e.g. Biocarta FAS Pathway) and also sets of targets regulated by a non-coding molecule (e.g. MIR3135B).

Each ome is processed, normalized, and analyzed according to its specific bioinformatics best practices (Section 1.1 in S1 Text). In case the ome is not natively represented as a

continuous numerical matrix, then StellarPath adopts peer-reviewed strategies. For instance, StellarPath accepts somatic mutation data in the form of a binary matrix where rows are genes, the columns are samples, and an entry of the matrix is either one or zero depending on whether the gene is mutated or not (Section 1.2 in S1 Text). The binary matrix is then transformed into a continuous numerical matrix using a network-based propagation algorithm (Section 1.3 in S1 Text), a method extensively proven effective for imputing missing values, identifying disease-associated mutated genes and detecting altered regulatory elements [25–28].

## Feature selection

StellarPath performs a comparative analysis of the patient's molecular profiles within each individual ome, mirroring a standard differential expression analysis. StellarPath detects molecules whose values (e.g. expression with genes, abundance with proteins, or other value types depending on the ome) show significant differences between the two classes.

Precisely, StellarPath computes two log fold changes for every molecule: expression log fold change (elFC) and variance log fold change (vlFC). Both measurements are the log2-transformed ratio between two coefficients that characterize the molecule's values across the two classes. The expression log fold change is determined with the coefficient being the average (ADV) [29,30] of the molecule's values in a class. In contrast, the variance log fold change uses the coefficient of variation (CV) [31] (the negated variance log fold change is also termed "stability"). For instance, when comparing cases versus controls, a molecule with negative elFC and positive vlFC (i.e. negative stability) is a molecule with consistently lower expression values in the cases compared to the controls.

StellarPath then fits the molecules to a linear model using both elFC and vlFC to identify those that are both highly differentially expressed and stable. The top-ranking molecules are selected based on criteria established by Smyth G.K. in linear models and empirical Bayes methods [32,33].

StellarPath combines the molecules selected from the different omics using a hierarchical-vertical integration flow. In other terms, it combines molecules from various analyses, such as differentially expressed genes and miRNAs, based on their annotated relationships. For instance, when considering these molecules, the method evaluates if changes in the expression of the non-coding molecules significantly correlate with alterations in the expression of their targets. One of the strategies employed by StellarPath consists in assessing the anti-correlation (or correlation, depending on the relationship) between a non-coding molecule (e.g., miRNA) and its target molecules (e.g., genes) using the empirical Cumulative Distribution Function (eCDF) (Section 1.1 in S1 Text).

StellarPath evaluates whether the sets of combined molecules (e.g. miRNA and its gene targets) significantly enrich pathways using an over-representation analysis [34]. It then categorizes the enriched pathways into four categories based on the prevalent types of expression and variance log fold changes observed in the significant molecules falling into these pathways. First, a pathway is considered activated (i.e. upregulated) by the case patients when it predominantly consists of molecules with positive expression log fold change and stability. Second, a pathway is considered activated by control patients when it exhibits a prevalence of molecules with negative elFC and negative stability. Third, a pathway is considered inhibited by the case patients when it is primarily composed of molecules with negative elFC and positive stability. Lastly, a pathway is considered inhibited by the control patients when it predominantly consists of molecules with positive elFC and negative stability (Section 1.4 in S1 Text).

## Patient similarity networks

StellarPath builds a patient similarity network per enriched pathway. It represents the training patients as nodes of a graph and adds a weighted edge for every pair of patients based on how similar they are. To understand the similarity measure used by StellarPath, let us consider a normalized count matrix $M$ (e.g. gene expression matrix) composed of patient's molecular profiles. In this matrix, rows correspond to distinct molecules, columns represent the $P$ patients and an entry $m_{ij}$ denotes the expression (i.e. abundance, methylation, or other measurements) of the $i\text{-}th$ molecule in the $j\text{-}th$ patient. Plus, let us also define a set called $PATHWAY = \{a_1, a_2, a_3, \ldots, a_n\}$ containing the row index of the significant molecules in $M$ that enriched a cellular function. Then the similarity measure employed by StellarPath evaluates two components (Section 1.5A in S1 Text).

The first component, also known as Stability Component, divides the smallest value that each molecule has in the two patients by its largest value. In other words, it assesses how much the same molecule has close values in the two patients (this reflects the variance lFC), the more the gap between the molecule's values is small and the more this component produces a high score:

$$Comp1_{PATHWAY}\left(P_j, P_k\right) = \frac{\sum_{i \in PATHWAY} min\left(m_{ij}, m_{ik}\right)}{\sum_{i \in PATHWAY} max\left(m_{ij}, m_{ik}\right)}$$

While, the second component, also known as Intensity Component, considers the average value in the two patients of each molecule, sums all the averages together, and divides the final amount by the number of significant molecules in the pathway. Dividing the sum by the number of significant molecules returns the average value of the molecule's average values between the two patients. In other words, this second component evaluates how much each molecule has high or low values in the two patents (this reflects the expression lFC). Precisely, If the pathway is activated, higher molecule's values contribute to a higher score:

$$Comp2_{PATHWAY}\left(P_j, P_k\right) = \frac{\left(\sum_{i \in PATHWAY}(m_{ij} + m_{ik})/2\right)}{|PATHWAY|}$$

Conversely, if the pathway is inhibited then StellarPath subtracts the second component from one such that lower molecule values contribute to a higher score:

$$Comp2_{PATHWAY}\left(P_j, P_k\right) = 1 - \frac{\left(\sum_{i \in PATHWAY}(m_{ij} + m_{ik})/2\right)}{|PATHWAY|}$$

Finally, the overall similarity between the two patients is calculated as follows:

$$Sim_{PATHWAY}\left(P_j, P_k\right) = \frac{\left(Comp1_{PATHWAY}\left(P_j, P_k\right) * Comp2_{PATHWAY}\left(P_j, P_k\right)\right)}{2}$$

The similarity measure calculates the area under the triangle defined by three vertices which are $(0,0)$, $(Comp1_{PATHWAY}, 0)$, and $(0, Comp2_{PATHWAY})$. This allows the two components to contribute equally, ensuring that the area (i.e., similarity) increases faster when both components increase than when only one component increases. This measure uniquely captures how the deregulated molecules modulate their pathway, assigning a high similarity only when the patients' molecules deregulate the cell's function in the same way. For example, two patients may have strongly expressed genes belonging to the EGFR pathway (size = 79) but discordant values for the same molecules; this may happen because, the disease of one patient is

acting on the pathway using EGFR and JAK [35], while the disease of the second patient is using IL-6 and GAB1 [36,37]. Traditional similarity measures such as the Pearson Correlation and the Euclidean distance do not capture this aspect (Section 1.6 and 1.7 in S1 Text) and would be mistaken in providing a high similarity value.

Once the PSNs are complete, StellarPath performs a second feature selection. It keeps only the networks where one patient class is cohesive (i.e. strong intra-similarities), the opposite class is sparse (i.e. weak intra-similarities) and the two classes are not similar (i.e. weak inter-similarities). To evaluate each PSN, a scoring system assigns it a value ranging from 0 to 10. The score is called Separability Power and increases based on the difference in cohesion between the classes; the higher the intra-similarities in the cohesive class compared to those in the opposite class and the inter-similarities, the higher the score. Precisely, StellarPath compares the intra-similarities of one class with those of the opposite class and the inter-similarities (Section 1.5C in S1 Text). It tests whether the low percentile (e.g. 0.35) of one class's intra-similarities is greater than the high percentile (e.g. 0.65) of the other two sets of similarities. In case this condition is satisfied, the PSN receives a score of 1. Next, the method iteratively adjusts the percentiles, decreasing the lower and increasing the higher, to potentially increase the score (Table A in S1 Tables). A PSN achieves the highest score of 10 when 90% (i.e. 0.1 percentile) of the similarities from one class is higher than 90% (i.e. 0.9 percentile) of the similarities of the other sets. From a biological point of view, the two patient classes are acting on the pathway differently; one class is cohesive because the members share a status leading to a stable (low variation between class members) deregulation (higher or lower expression compared to the opposite class) of the pathway, while the opposite class is composed by dissimilar patients because either their shared status is not requiring the pathway or there is a secondary biological factor differentiating the patients (e.g. disease subtypes, misdiagnosis and comorbidities) [38].

The PSNs with a power lower than 1 are filtered out, while the networks with a power higher than 0 are called signatures of the cohesive class and participate in the downstream analysis.

## Network processing

StellarPath analyses the class-signature pathway-specific PSNs to infer new information. It determines how much each patient is similar to the members of its class and dissimilar to the non-members for every network.

Let us define $CL = \{c_1, c_2\}$ where $C_1$ contains all the indexes pointing to the patients of one class and $C_2$ the remaining ones, then the vector of the similarities of each patient is defined as follows:

$$S = \left\{ s_{1j}, s_{2j}, \ldots, s_{(|c_1|+|c_2|)j} \mid \forall j \in \{c_1 \cup c_2\} \right\}$$

where $S_{1j}$ is the vector of the similarities of the patient with index $i = 1$ with any other patient in a PSN. StellarPath measures the centrality of a patient between the classes as follows:

$$centrality(i) = \begin{cases} if\ i \in c_1\ then\ \dfrac{(\ Median(s_{ij} \mid \forall j \in c_1)\ *\ (1 - Med(s_{ij} \mid \forall j \in c_2)\ )}{2} \\[2ex] if\ i \in c_2\ then\ \dfrac{(\ Median(s_{ij} \mid \forall j \in c_2)\ *\ (1 - Med(s_{ij} \mid \forall j \in c_1)\ )}{2} \end{cases}$$

The centrality values of the patients are then scaled between zero and one to make them comparable between PSNs; the patient who is the most similar to the members of its own class

and most dissimilar to the other patients will always get a value equal to one, while zero in the opposite case. The component $Median(s_{ij}|\forall j \in c_1)$ calculates the median of $i$-th patient's similarities with the patients in $c_1$. Component $1—Median(s_{ij}|\forall j \in c_2)$ captures the median of $i$-th patient's dissimilarities with the patients in $c_2$. While the division by two in the formula is part of the calculation for the area under the triangle defined by three vertices: $(0,0)$, $(Median(s_{ij}|\forall j \in c_1), 0)$ and $(0, (1- Median(s_{ij}|\forall j \in c_2))$.

The centrality value per pair of patient and pathway enables the resolution of the stratification task, in fact, pathway-specific PSNs which are signature of class $c_1$ can be further divided into subsets where each one is assigned to the subgroup of its most central patients.

After the centrality computation, StellarPath performs a sparsification of each signature PSN. It removes the lowest similarities of each patient; in this way, the network's topology reflects the similarities, the members of the cohesive class are quite connected, while the opposite class loses most of its connections. The same rule will be applied and satisfied also when the testing patients are added to the network together with their connections to the training individuals.

## Patient classification

StellarPath trains a graph convolutional network with each PSN. The primary role of the GCN is to understand how patients are connected, identify the topological characteristics that differentiate the two classes' members, learn rules to accurately determine the class of known patients, and apply the rules to predict the class of unseen patients.

GCN is a deep learning architecture designed to learn a meaningful representation, called embedding, for each patient (node) in the PSN and establish how to assemble this representation such that reflects the patient's class (node's label). The embedding can be seen as a compact way (low-dimensional vector) to describe an object and here is designed to encode relevant information about the patient's role, relationships, and context within the PSN.

For clarity, we categorize patients as "known" when their class is recognized by the GCN, "unknown" when the GCN is unaware of their class and does not use this information, and "unseen" when they have not been previously evaluated by the GCN. Thus, patients in the training set are known, those in the validation set remain unseen until they are used by the GCN, and patients in the testing set are entirely unknown.

During the training phase, the GCN operates on its input PSN, which includes only known patients, to learn how to assemble their embeddings by aggregating information from their neighbors, such as similarities and labels. This process involves assigning importance to each piece of information using learnable parameters (as if patient A is more useful to predict other patient's classes than another patient B, so the GCN assigns a higher weight to A than to B). By refining these parameters, the GCN aims to capture key characteristics that distinguish the different classes and generate meaningful embeddings. Here the GCN is called to learn from the training patients and minimize the error of misrepresenting the validation patients in relation to their true class. The minimization is conducted by adjusting the learnable parameters.

After the training phase, the GCN applies the learned knowledge to the same PSN, now including unknown testing patients. It generates embeddings for these unseen individuals and subsequently translates their embeddings into class labels, categorizing them into one of the two classes.

For the training, let a PSN be defined as a weighted graph $G = (V, E, W)$ where $V = \{v_1, v_2,...,v_n\}$ is a set of nodes, $E \subseteq V \times V$ is a set of edges between nodes in $V$ and $W$ is the adjacency matrix mathematically defined as an $|V|x|V|$ matrix. Each entry $w_{ij}$ in $W$ corresponds to the connection and similarity $Sim_{PATHWAY}(P_i, P_j)$ between the $i$-th and the $j$-th patient nodes. The

similarity is calculated with the significant molecules enriching a pathway, so the graph could be denoted with $G_{PATHWAY}$.

The nodes of the PSN represent known training patients who are associated with a class (i.e. text label), so let *CLASSES* = *{$CL_1$, $CL_2$}* be the set of class labels and $y_i$ be the class label associated with the *i-th* patient such that $y_i \in$ *CLASSES* and $y_i \in y = \{y_1, y_2, \ldots, y_n\}$. This means that the graph is augmented with nodes' attribute vectors $X = \{x_1, x_2, \ldots, x_n\}$, where $x_i$ describes the $v_i$ node. StellarPath performs a binary classification and does not use other categorical information to describe the patients, so $x_i$ is simply binary encoding the class ($c_1$ $c_2$ have been previously defined for the centrality measure):

$$x_i = \begin{cases} if\ i \in c_1\ then\ 0 \\ if\ i \in c_2\ then\ 1 \end{cases}$$

Finally, StellarPath feeds the graph's information to a GCN called GraphSage [39] that learns how to predict the patient nodes' labels based on the information of the neighborhoods (Section 1.8 in S1 Text).

Once the model has been trained and optimized to predict the training and validation patients at the best, StellarPath introduces unknown patients into the PSN and calls the model to predict their class.

As overview, StellarPath trains and applies a GCN on each pathway-specific PSN to predict the class of unknown patients. In the training phase, the embedding functions learn to aggregate information from the training patients, aiming to accurately predict their class. To assess the effectiveness of this training, StellarPath introduces the validation patients. These patients, unseen but known, serve as a benchmark for the GCNs. Only when a GCN proves proficiency in predicting the class of the validation patients, it is used to predict the class of entirely unknown patients, whether they are part of the cross-validation testing set or part of an external dataset in the inference phase. This process is reiterated for each significant PSN. Each PSN provides a class prediction for an unknown patient. The final class prediction for an unknown patient is then determined by majority voting, assigning the class that has been most frequently predicted across all PSNs.

StellarPath's majority vote approach is employed also by traditional machine learning algorithms as Random Forest (RF) classifiers. Both StellarPath and RF classifiers work on the ensemble learning principle, where multiple models are trained on different data and their collective predictions are used to make the final decision. In the case of StellarPath, the models are GCNs trained on different pathway-specific PSNs. Each PSN reflects how patients are similar according to how they regulate a pathway. Similarly, a RF classifier consists of multiple decision trees. Each tree is trained on a different subset of the available data. The final class prediction in both methods is determined by a majority voting system. The class that is most frequently predicted across all GCNs (in the case of StellarPath) or all decision trees (in the case of RF) is assigned as the final predicted class. StellarPath implemented the classification with GCNs in Python (Section 1.1 in S1 Text).

## Output

StellarPath provides the predicted classes and the significant PSNs together with their information such as cohesive class, enriched pathway, expression and variance fold change of the significantly deregulated molecules, separability power, the average of the intra and inter-similarities of the classes, and the centrality score for each patient.

The PSNs can be visualized with an ad-hoc implementation that reduces the number of patients who are visible as nodes without losing the information about the invisible ones; this

helps to reduce the complexity of the PSN and improves the interpretability of the patient similarities, especially in case of hundreds of patients. The idea behind is to group up the strongly similar patients and select one member of each group to represent the excluded ones. Given a PSN, StellarPath determines how much two patients are similar based on how much they are similar to the others with the following formula:

$$Sim2_{PATHWAY}\left(P_i, P_j\right) = \frac{\sum_{d \in CL} min\left(w_{di}, w_{dj}\right)}{\sum_{d \in CL} max\left(w_{di}, w_{dj}\right)}$$

The measure uses the operator $d$ to iterate over the rows of the adjacency matrix $W$ describing the similarities owned by the patients $P_i$ and $P_j$ in a pathway-specific PSN. The latter measure is used to create the network $PSN^2$ which does not reflect how much two patients are similar based on their molecules' expressions anymore, but it reflects how much the patients have similar connections with others in a pre-existing class-signature pathway-specific PSN. Next, the adjacency matrix of the $PSN^2$ is column split in two such that each submatrix contains only the patients belonging to one class and undergoes an unsupervised clustering performed with the K-means Lloyd's algorithm [40]. The patients of every submatrix are divided into a certain number of clusters (the default number is 20 but can be personalized by the user, a higher number will lead to a more authentic representation of the original PSN while a lower number will decrease the complexity) and the patients closer to the centroids are selected to represent the other ones in the original PSN. Lastly, the position of the patient nodes in the plot is determined using Fruchterman & Reingold's force-directed layout [41]. The network becomes easy to analyze and still representative of how all the patients are similar.

## Precision medicine

StellarPath offers predictive, pathway-specific patient similarity networks associated with deregulated molecules, providing insights into patient similarities. This facilitates multiple analytical operations with significant impacts on precision medicine.

It enables the development of unique treatments for patients who show dissimilarities in key disease pathways compared to most of their class. For instance, if the Oxidative Stress pathway is upregulated in lung cancer patients [42] as opposed to controls, those patients with lower centrality in the PSN (indicating greater similarity to controls) may have a less advanced stage of cancer. Consequently, these patients would benefit from a more personalized treatment.

It allows the estimation of a treatment's effectiveness over time. By sequencing lung cancer patients before and during therapy, we can then compare the PSNs related to the same pathways at different times. This comparison can highlight changes in network structure and patient centrality, reflecting the molecular impact of the therapy. For example, the Oxidative Stress specific PSN might indicate a patient's response to treatment through a reduced centrality over time, suggesting regulation of the pathway akin to that of the control individuals. Additionally, this approach can pinpoint which deregulated molecules influence the efficacy of the treatment.

StellarPath's trained model and predictive PSNs can be used to predict the disease class of new patients. This capability boosts the diagnostic process and treatment planning, enabling faster and more robust medical decisions.

Finally, with StellarPath, it is possible to stratify patients within the same class based on their centrality in a PSN and to test if this stratification is consistent across multiple pathways. Identifying a consistent pattern of patient subgroups could lead to the division of the pathways

for the functional characterization of disease subtypes, with each subtype potentially predominated by a specific patient subclass. Alternatively, each subclass could be correlated with a category of clinical information. In either scenario, for diseases like lung cancer, this could facilitate the development of treatments for each disease subtype or the integration of correlated clinical information as a covariate in drug or treatment design.

# Results

## Classification performances

We tested the ability of StellarPath to classify with sixteen datasets (Table B in S1 Tables). Precisely, fifteen publicly available TCGA datasets were obtained using the R packages: TCGAutils [43], curatedTCGAData [44] and RTCGAToolbox [45]. While one dataset was obtained by sequencing cortical neurons of wild-type (WT) mice [46]. Each TCGA dataset included information on the patient's somatic mutations and read counts of genes, long non-coding RNAs (lncRNAs), and micro RNAs (miRNAs). We kept only patients characterized by a primary tumor sample and their tumor stage. We then categorized these stages into Early (stage I or II) or Late (stage III or IV) based on the tumor/node/metastasis (TNM) system [47–49]. Meanwhile, our WT dataset included read counts of genes, long non-coding RNAs (lncRNAs), and micro RNAs (miRNAs) of samples treated for 12h with oxygen and glucose deprivation (OGD WT) compared to the normoxic condition (N WT). StellarPath's objective was to classify TCGA patients into Early or Late stages and our mouse samples into OGD WT or N WT classes. These datasets were also used to test netDx [11]. Unlike StellarPath, netDx is not a pathway-based classifier but is designed to generically handle any datum as long as the user implements the data processing and the similarity measure for building the networks. For our purposes, we configured netDx to generate pathway-specific PSNs using its default similarity measure, the Pearson Correlation, for sequencing data.

We used both methods to classify the patient classes from each dataset, employing a five-fold cross-validation approach. Both methods were supplied with data on patient classes, omics, canonical pathways from MSigDB [19], GO [20], and Kegg [21], regulatory networks of miRNAs and lncRNAs from miRWalk [22], and RNAInter [23] databases. As a result, we obtained the predictions of the testing patients and the best pathway-specific PSNs selected by the two methods to classify each dataset. We assessed the classification performances (Table C in S1 Tables) by measuring the Matthews Correlation Coefficient (MCC) [50,51] which compares the predictions with the patients' true class (Fig 2) (Section 1.9 in S1 Text).

StellarPath's GCNs, trained on pathway-specific PSNs (Table C in S1 Tables), performed well across all datasets. The MCC values estimated by comparing the predictions of every GCN with the patient's true classes consistently exceeded 0.7, with accuracy values surpassing 0.85, sensitivity values over 0.93, and specificity values above 0.74. As expected from a deep learning model, the performance of the GCNs improved with increasing dataset size. However, some GCNs struggled to correctly classify testing patients when trained on very small PSNs, such as those with only ten samples from the OGD dataset. Despite a few GCNs underperforming in each dataset, most of them achieved an MCC value over 0.7. As a result, when StellarPath used the Ensemble strategy and combined the GCNs predictions, it consistently predicted the correct class for each test patient across all datasets. Specifically, StellarPath ascertained the final class of each testing patient with nearly 80% of the GCNs concurring in their predictions across all datasets (Table D in S1 Tables). In contrast, netDx's performance varied, with MCC values ranging between 0.5 and 0.75. In summary, StellarPath outperformed netDx in every dataset and was even capable of handling the small sample size of the OGD dataset.

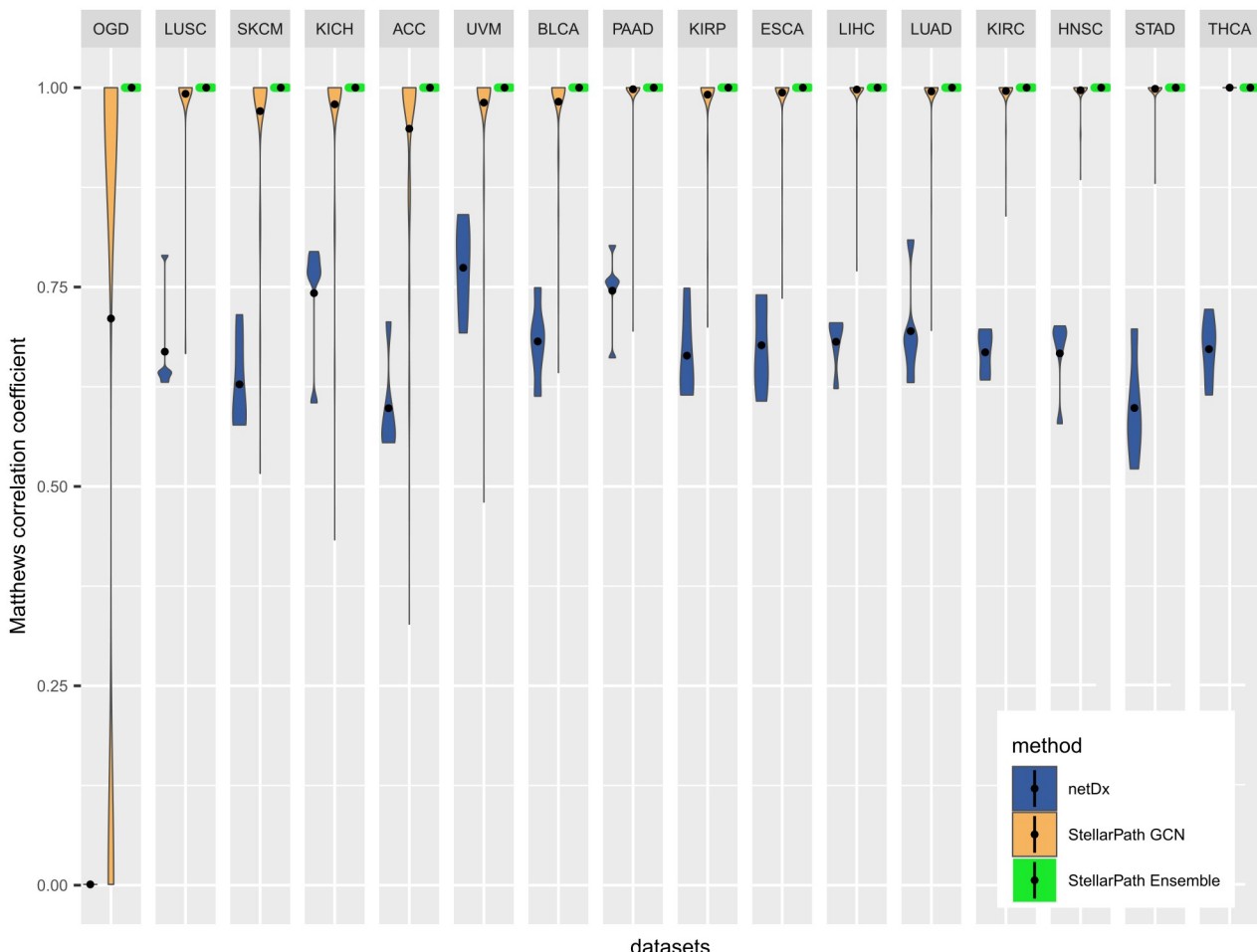

**Fig 2. Classification performances.** Violin plot of the classification performances (Table C in S1 Tables) measured with the Matthews Correlation Coefficient (MCC). For netDx, one value of MCC is determined with the predictions made by the classifier applied to one non-pathway consensus PSN for each run of cross-validation. For StellarPath, there are two types of violins. One value of MCC composing a StellarPath GCN violin is determined with the predictions made by a GCN using one pathway-specific PSN. One value of MCC composing a StellarPath Ensemble violin is determined with the predictions made by majority voting from all the trained GCNs. The dot of a violin represents its average.

The performances achieved by StellarPath using GraphSAGE align well with the ones observed in other applications of GCNs [52–54]. The pathways have also proven to effectively work as features for challenging patient classification tasks as previously shown by Rahimi et al. [55]. Finally, the outcomes from the ensemble strategy are optimal, but this is expected since the GCNs were trained and tested on PSNs with selected meaningful topologies, representing pathways and molecules that already distinguished patient classes based on their inherent qualities. The most important insight is actually whether a GCN found a PSN useful for the prediction. In this case, the PSN would be pathway-specific class-signature and predictive.

## Predictive pathways

StellarPath and netDx provide pathways as outcomes of their classification processes, but they determine which pathways to present to the user based on distinct criteria. Precisely, netDx returns the pathways (Table E in S1 Tables) considered predictive in the seventy percent of

cross-validation runs. In contrast, StellarPath adopts another approach. After the classification, StellarPath assigns the predicted classes to the previously unknown patients and then re-analyses the omics, incorporating their molecular profiles. In this way, StellarPath identifies deregulated molecules, enriched pathways, and class-signature PSNs that remain significant even when considering the data from these newly classified patients. Finally, it returns the pathways (Table F in S1 Tables) used to classify and that maintained their significance when the predicted patients were included in. StellarPath's resulting markers are important for classifying and biologically meaningful for both training and predicted patients. We compared the amount and the types of pathways that StellarPath and netDx provide as a result of their classification in the tables below (Table 1).

The number of pathways identified by StellarPath's analysis tends to decrease with increasing dataset size. This is because, as the dataset grows, it becomes more challenging for StellarPath to identify a class that is more cohesive than the opposite one within the PSNs, resulting in a separability power greater than zero. In contrast, the number of pathways identified by netDx mostly increases with dataset size, as larger datasets make it easier for netDx to find a PSN capable of predicting patient classes.

About the types of pathways, StellarPath found many significant sets of miRNA's targets with valid anti-correlated relationships and one significant set of lncRNA's targets. On the opposite, netDx did not find predictive pathways with the information coming from the miRNAs expression. About somatic mutation data, StellarPath found few predictive and enriched gene canonical pathways that present a mutated gene. While netDx found many mutated pathways. This reason is due to the different ways the two methods use mutational information. StellarPath finds genes that are more likely to be mutated in one class than in the other and then it checks whether these genes belong to deregulated pathways found with the analysis of transcriptomics data. Instead, netDx does not look for overlaps but for gene canonical pathways "enriched" by the significantly different mutated genes. This specific result highlights the benefit of a parallel integration flow, which does not need to align the results from different omics analyses. However, a drawback is that a pathway found significant based on mutational information might not be significant when evaluated with other omics, potentially leading to contradictions or paradoxes.

Finally, we researched the pathways uniquely found by StellarPath in classifying three TCGA datasets. StellarPath identified pathways that are enriched and upregulated in late-stage cancer patients, marking the transition from the early stages of lung squamous cell carcinoma (LUSC). The pathway of Negative Regulation of Immune Response enhances the tumor's

**Table 1. StellarPath and netDx predictive pathways.** Type and number of significant and predictive pathways resulting from StellarPath analysis and netDx analysis.

| Method:Pathway type/Dataset | OGD | LUSC | SKCM | KICH | ACC | UVM | BLCA | PAAD | KIRP | ESCA | LIHC | LUAD | KIRC | HNSC | STAD | THCA |
|---|---|---|---|---|---|---|---|---|---|---|---|---|---|---|---|---|
| StellarPath:non-mutated gene canonical pathway | 117 | 97 | 84 | 73 | 231 | 59 | 78 | 135 | 96 | 65 | 28 | 59 | 23 | 32 | 7 | 55 |
| StellarPath:mutated gene canonical pathway | 0 | 5 | 1 | 2 | 19 | 0 | 0 | 4 | 0 | 0 | 3 | 0 | 0 | 0 | 0 | 0 |
| StellarPath:non-mutated lncRNA target set | 0 | 0 | 0 | 0 | 1 | 0 | 0 | 0 | 0 | 0 | 0 | 0 | 0 | 0 | 0 | 0 |
| StellarPath:non-mutated miRNA target set | 156 | 65 | 27 | 27 | 66 | 43 | 31 | 36 | 21 | 22 | 32 | 25 | 22 | 20 | 6 | 10 |
| StellarPath:mutated miRNA target set | 0 | 1 | 2 | 6 | 18 | 0 | 0 | 8 | 0 | 0 | 0 | 0 | 0 | 0 | 0 | 0 |
| netDx:mutational pathways | 0 | 80 | 84 | 116 | 74 | 71 | 173 | 304 | 105 | 203 | 223 | 143 | 178 | 211 | 174 | 79 |
| netDx:gene canonical pathways | 0 | 14 | 14 | 1 | 140 | 7 | 24 | 164 | 108 | 53 | 38 | 161 | 34 | 95 | 28 | 50 |

ability to evade the immune system, contributing to its survival [56]. Myeloid Activation within the tumor microenvironment increases the aggressiveness [57,58] of the cancer. Aberrant STAT5 signaling leads to the overexpression of genes that results in increased cell proliferation, migration, invasion, and disrupted immune surveillance [59]. The upregulation of the Cell-Cell Adhesion pathway enables cancer cells to detach from the primary tumor and form metastases at distant sites [60]. Lastly, and most importantly, StellarPath detected a deregulation in hsa-miR-551b and its gene targets. This microRNA is identified as a key microRNA associated with poor prognosis, tumor progression, and the late cancer stage in lung adenocarcinoma [61]. Evidence supporting the significance of pathways enriched and predictive of late-stage cancer patients within the skin melanoma (SKCM) and pancreatic adenocarcinoma (PAAD) datasets has also been identified (Section 1.10 in S1 Text).

## Topology of patient similarity networks

StellarPath and netDx both provide PSNs representing pathways from their classification outcomes, yet their selection criteria diverge. netDx selects PSNs of pathways that are predictive in seventy percent of cross-validation runs, whereas StellarPath chooses those pathway-specific PSNs that best differentiate the classes topologically. This suggests that while netDx's PSNs are effective in classifying, they may not always have a biologically meaningful pattern. On the other hand, StellarPath primarily aims to detect PSNs that convey biologically relevant information. To illustrate, we collected the resulting PSNs from both methods and compared them (Fig 3) using two metrics, the Separability Power (Methods "Patient Similarity Networks") and the Jaccard Index. We used the Power to measure how much each PSN includes one cohesive class, one sparse class, and low similarities between the two classes. We used the Jaccard Index to measure how much the PSNs correctly separate the two classes, independently of their cohesiveness or sparsity. Precisely, given a PSN, we profiled each patient with its centrality (Methods "Network Processing"), performed unsupervised clustering (S1 File), and measured the overlap between the PSN's node clusters and the original patient classes (Section 1.11 in S1 Text).

StellarPath classified each multi-omics dataset using PSNs that exhibit higher Power values compared to the predictive networks of netDx. Additionally, we applied the Jaccard Index to assess the extent to which each PSN separated the patient classes. For instance, the Jaccard Index equal to one is achieved when a PSN has classes that are both cohesive within and distinct from each other. StellarPath outperformed netDx also with this metric. StellarPath shows the highest Jaccard Index in thirteen datasets out of sixteen. Consequently, StellarPath's PSNs provide a clearer distinction between patient classes and are easier to interpret for further analysis. For instance, this means that the end user can easily confirm the correctness of the method's feature selection. Most importantly, it ensures that the selected PSNs are biologically relevant, trustworthy, and can clarify why a specific pathway is considered significant.

## Biological relevance

StellarPath presents itself as a classifier designed to select, use, and provide markers that are both predictive and biologically meaningful, aiding in the comparison and characterization of the studied classes. Given this objective, we validated the markers found by StellarPath. Precisely, we examined the significant upregulated molecules and pathways that StellarPath selected and used to build the predictive OGD WT signature PSNs (i.e. networks where OGD WT samples were more similar than N WT samples and the two classes were dissimilar). We performed a functional enrichment analysis using QIAGEN IPA (QIAGEN Inc., https://digitalinsights.qiagen.com/IPA) [62]. This external software exploits a manually curated

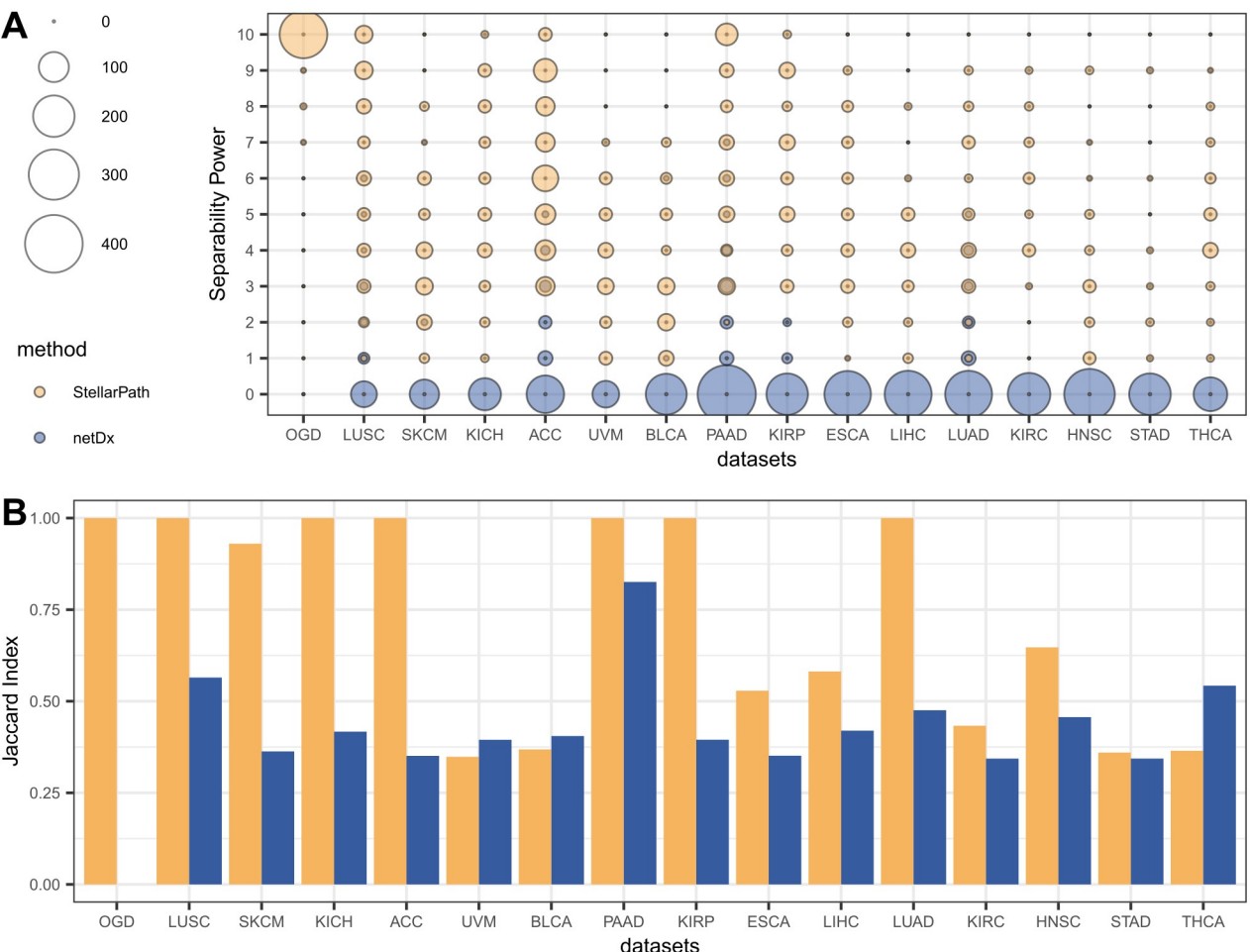

**Fig 3. Network topology.** (A) Scatter plot comparing the separability of the patient's classes in the PSNs selected by StellarPath and netDx for classifying the datasets. The separability of the classes is assessed with StellarPath's Power ranking system. Each dot's size indicates the number of PSNs with the corresponding Power value on the Y-axis. The dot's color signifies which method has a greater number of PSNs with a specific Power. (B) Bar plot comparing the Jaccard index about the quality of the unsupervised patient clusters identified in the PSNs compared to the real patient's classes of each dataset. The Y-axis refers to the Jaccard index. A taller bar indicates a closer overlap between the unsupervised patient clusters and the actual patient classes.

database of molecule-phenotype associations. We used it to find the subset of deregulated molecules under examination which were annotated and associated with Hypoxia and Disorder of Glucose Metabolism (Section 1.12 in S1 Text and Table G in S1 Tables). Finally, we linked the significant pathways under examination to the same phenotypes whether their sets contained annotated deregulated molecules. The insights derived from combining StellarPath and IPA are illustrated in Fig 4.

The IPA analysis revealed that StellarPath selected many deregulated pathways and molecules that are correctly associated with Hypoxia and the Disorder of Glucose Metabolism. Notably, the statistics employed by StellarPath to rank and prioritize biological markers appear to align with their biological relevance. After the annotation, it becomes evident that Oxygen-Glucose Deprivation (OGD) impacts multiple pathways. According to StellarPath's findings, cells in OGD WT samples upregulate "Hallmark Hypoxia" and "PID HIF1 TFPATHWAY" to cope with reduced oxygen levels. At the same time, these cells begin to rely more on glycolysis,

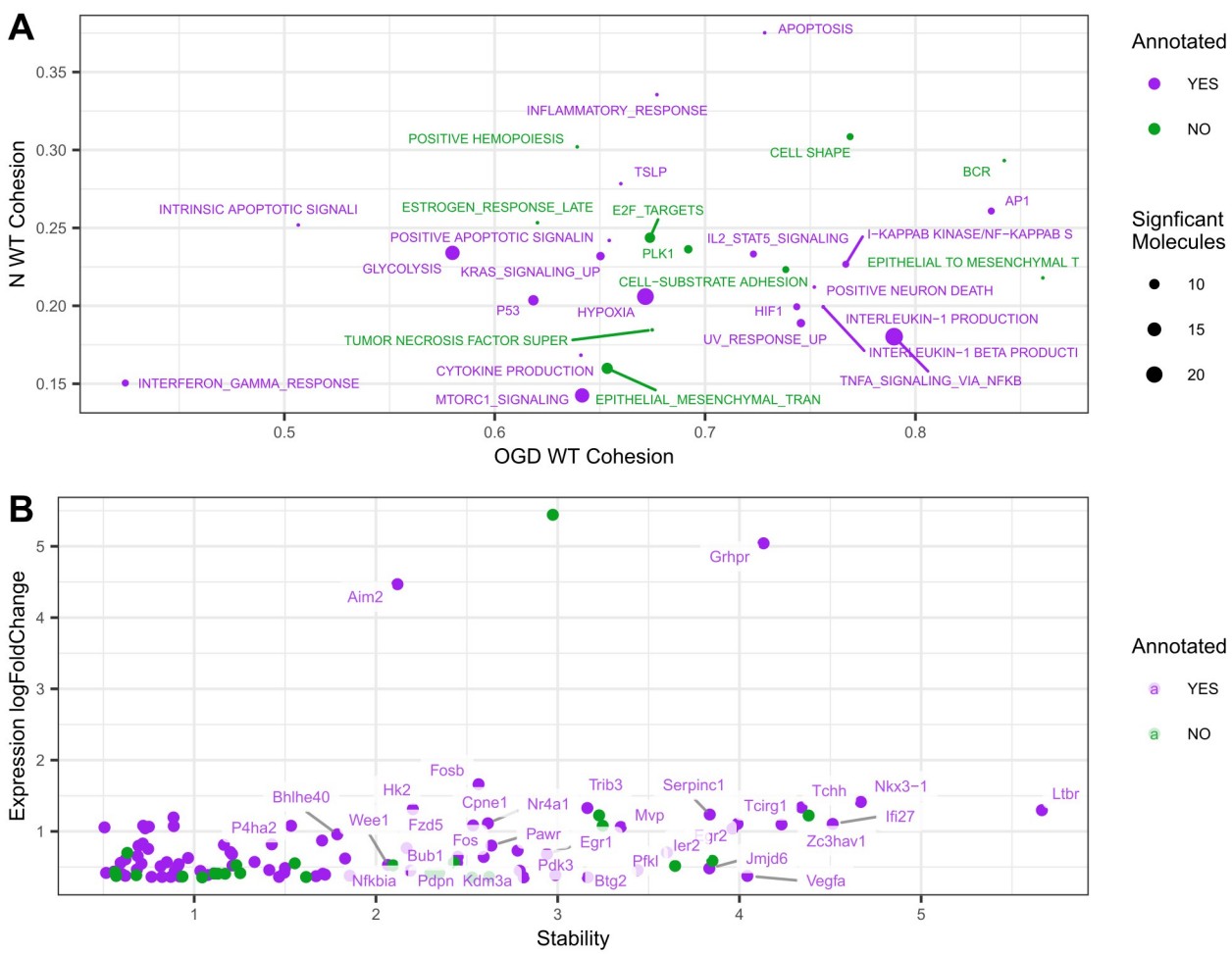

**Fig 4. Molecules and pathways relevance.** (A) Scatter plot of the pathways represented by predictive OGD WT signature PSNs. The X-axis indicates how much the OGD WT samples are cohesive in the PSN associated with a pathway. The Y-axis indicates how much the N WT samples are cohesive. The size of the dot indicates how many molecules are significantly deregulating a pathway. The color indicates if a pathway is associated with the OGD phenotype based on IPA. The best pathways are in the bottom right corner because of strong similarities between OGD WT samples and weak similarities between the N WT ones. (B) Scatter plot of the deregulated molecules belonging to the enriched pathways represented by predictive OGD WT signature PSNs. The X-axis indicates how much a molecule is stable in the OGD WT samples against the controls. The Y-axis indicates how much a molecule is deregulated between OGD WT versus N WT. The best molecules are in the top right part of the plot because they are the most deregulated and stable in OGD WT.

as indicated by the activation of the "Hallmark Glycolysis" pathway. The combined deprivation of both oxygen and glucose induces cellular stress, leading cells to regulate cell death and inflammation. This is evident from the upregulation of pathways such as "Hallmark P53 Pathway", "Hallmark Apoptosis", and "Hallmark Inflammatory Response". Overall, these upregulated annotated pathways describe well how cells respond to not having enough oxygen and glucose.

Furthermore, we exploited the samples' centralities (S1 File) within the pathway-specific predictive PSNs to provide examples of biological and clinical implications. For instance, the 'OGD_WT.4' sample shows the lowest centrality in the 'Hallmark Hypoxia' pathway among the OGD samples, indicating that its gene regulation is more akin to control samples than to the majority of those under Oxygen-Glucose deprivation. This suggests that 'OGD_WT.4' might not have been significantly affected by the deprivation treatment, marking it as a

potential outlier. A wider investigation reveals two subclasses within the OGD samples, particularly in pathways related to immune response, inflammation, and cell death ("Hallmark Interferon Gamma Response", "Hallmark IL2 STAT5 Signaling", "TSLP" and "Positive Regulation of Apoptotic Signaling Pathway"). The 'OGD_WT' sample, like 'OGD_WT.4', displays lower centrality in these pathways than the rest of the class, suggesting a delayed or different response to the treatment.

These insights are particularly valuable for biologists or clinicians who aim to understand the nuances of each sample's response and the inter-sample dynamics. Given that these samples are derived from in-vivo mouse models, which are inherently more complex than in-vitro models, such results from StellarPath can be useful in deciphering the biological uniqueness of each sample. For instance, further studies could explore the phenotypic characteristics of "outlier" models like 'OGD_WT' and 'OGD_WT.4' to understand their relative resilience to the treatment. Moreover, investigating genotypes and single nucleotide polymorphisms in these OGD subclasses could allow to determine markers that impact the treatment susceptibility. Finally, StellarPath's results could guide pilot studies (e.g. targeted RNA sequencing) in selecting appropriate models, subsequently refining the focus for bigger and more comprehensive studies (e.g. increasing the power and doing total RNA sequencing) based on cohesive treatment responses.

## Comparison with GSEA

StellarPath's main results are enriched pathways represented by class-signature and predictive PSNs. These pathways reflect the molecular differences between the patient classes under study. From this perspective, StellarPath resembles traditional functional enrichment analysis tools. Thus, we investigated the differences and similarities between the pathways provided by StellarPath and the Gene Set Enrichment Analysis tool (GSEA) [19].

We applied StellarPath on our mouse dataset where the classes in comparison are Oxygen-Glucose deprived Wild-Type and Normoxia Wild-Type. In parallel, we applied the Gene Set Enrichment Analysis tool (GSEA Broad Institute algorithm) implemented in the "fgsea" R package [63] on Bioconductor. Since GSEA does not work on multi-omics data, we used it only on the gene expression matrix, normalized with Limma workflow. Precisely, we applied GSEA in two modalities called Label-permuting and Preranked, and we kept the resulting pathways with an adjusted probability value lower than the 0.05 standard threshold. Lastly, we compared the number of pathways presented by each method in the table below (Table 2) and identified which pathways were found either uniquely or shared between the methods (Section 1.13 in S1 Text).

Preranked GSEA found the highest number of significant pathways, Label-permuting did not find any, while StellarPath detected sixty-eight. These numbers reflect the methods. Label-permuting is the most conservative method because it checks if the pre-shuffling enrichment

**Table 2. Number of significant pathways found by StellarPath and GSEA.**

| Methods | Number of Pathways |
|---|---|
| StellarPath | 68 |
| Label-permuting GSEA | 0 |
| Preranked GSEA | 481 |
| Shared by StellarPath and Preranked GSEA | 34 |
| Unique StellarPath | 34 |
| Unique Preranked GSEA | 447 |

score (ES) of a pathway is significant compared to the ESs obtained after shuffling the samples' classes. Preranked GSEA is the least conservative employing a gene label permutation for its null distribution of ESs. Meanwhile, StellarPath demands that pathways meet multiple aligned conditions, such as including differentially expressed molecules, be over-represented, be represented by a class-signature PSN where the classes are separated, and produce a predictive PSN that can classify the patients.

When scrutinizing the pathways, a large fraction of those shared (Table H in S1 Tables) between Preranked GSEA and StellarPath are tied to Oxygen-Glucose deprivation. Hypoxia-related pathways are directly linked to low oxygen conditions, metabolic pathways align with cellular responses to energy stress, while inflammation and apoptosis pathways reflect cellular stress and injury responses. About distinct findings, StellarPath exclusively found inflammatory pathways (Table I in S1 Tables) activated in OGD WT samples like "Hallmark Inflammatory Response", and homeostatic pathways activated in N WT samples like "Cellular Transition Metal Ion Homeostasis". On the contrary, Preranked GSEA identified pathways (Table H in S1 Tables) both pertinent to OGD WT, like "Cellular Response to Oxygen Levels", and more general pathways about muscle development, organ morphogenesis, and ion transport.

In conclusion, GSEA finds many more pathways than StellarPath, some of them are connected to OGD but many of them are generic or unrelated (Section 1.14 in S1 Text) to the classes in comparison. Plus, GSEA does not provide further information to understand if a pathway is significantly deregulated by one specific class. Instead, StellarPath demonstrates a stronger performance in identifying pathways relevant to Oxygen-Glucose deprivation. Compared to GSEA, StellarPath focuses on pertinent pathways, minimizes the inclusion of unrelated ones, and can recognize that the most generic ones (i.e. homeostatic) are regulated by the Normoxia samples and not by the OGD samples which are more oriented to handle the induced stress.

## Inference

StellarPath is not only designed for the identification of molecular markers and classification of patients in a cross-validation setup. It is also capable of re-using the learned markers to infer the classes of completely new patients. This inference capability is beneficial when one has trained and tested StellarPath on a known dataset and wishes to employ the identified predictive molecules, pathways, and PSNs to classify patients from an external independent dataset.

We show that StellarPath can solve an inference task exploiting four publicly available RNA sequencing datasets [64–67] describing Chronic Lymphocytic Leukemia (CLL) patients divided into two subtypes (Table J in S1 Tables). Precisely, CLL is a type of cancer that can be divided into two subtypes based on the frequency of somatic hypermutation (SHM) in the immunoglobulin heavy chain V (IGHV) region: unmutated CLL (UM-CLL) and mutated CLL (M-CLL). The distinction between these subtypes is crucial as they respond differently to treatments and have varied prognoses. The four datasets have been produced for different studies. Thus, firstly, we trained StellarPath on the biggest dataset to identify deregulated molecules, enriched pathways, and predictive pathway-specific patient similarity networks (PSNs) between UM-CLL and M-CLL patients (Table K in S1 Tables). Secondly, we applied the trained StellarPath to classify UM-CLL and M-CLL patients in the other three independent datasets. As last, we retrieved information about the predictive pathway-specific PSNs that StellarPath found during the training and used to classify the new patients (Fig 5A), and compared the predicted classes of the unknown patients with their true subtype to assess the quality of the inference (Fig 5B) (Table L in S1 Tables).

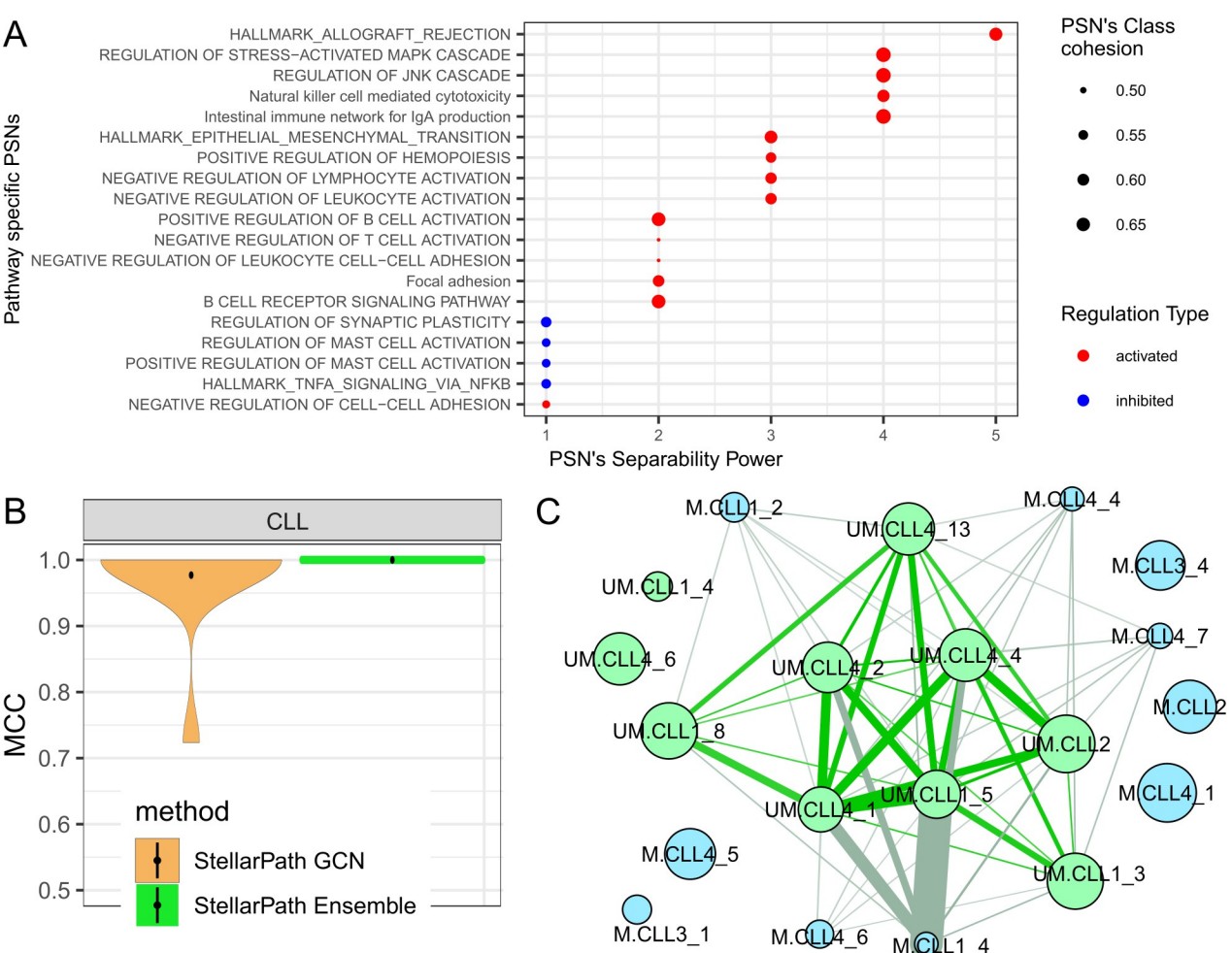

**Fig 5. Inference.** (A) Scatter plot of the pathways represented by predictive UM-CLL signature PSNs that StellarPath found during training. The X-axis indicates the rank of separability between the two classes. The Y-axis indicates the name of the pathways. The size of the dot depends on the smallest similarity (low percentile of the Separability Power system) between UM-CLL patients which is higher than the highest similarity (high percentile) between M-CLL patients. The color indicates how the UM-CLL subtype is deregulating the pathway. (B) Violin plot of the classification performances assessed comparing the predicted classes of the unknown patients to their true subtype. (C) UM-CLL signature PSN of the activated B Cell Receptor Signaling Pathway of Power 2, where UM-CLL patients are represented by green nodes, M-CLL patients by blue nodes, green lines represent edges between UM-CLL nodes, grey lines represent inter-similarities and blue lines are between M-CLL nodes. A line thickness represents the similarity value associated with it. The size of the node is determined by the patient's centrality. The PSN is sparse because the edges with low similarity have been hidden. Thanks to StellarPath's plot function, the PSN represents 123 patients in 20 nodes.

In the training and testing phases, StellarPath identified pathways that were either activated or inhibited, essentially deregulated, only by the UM-CLL subtype. Following cross-validation, StellarPath used these pathway-specific PSNs to classify patients from the previously untouched datasets. All the generated GCNs predicted the classes of these new patients with Matthews Correlation Coefficients (MCC) above 0.7. Subsequently, StellarPath aggregated these predictions, assigning to each unknown patient the class that received the majority of votes. In detail, the UM-CLL signature PSNs allowed the GCNs to unanimously predict the classes of the unknown UM-CLL patients, however, ten percent of them voted wrongly the class of the M-CLL patients (Table M in S1 Tables).

In the context of Chronic Lymphocytic Leukemia (CLL), StellarPath's resulting pathways can find common ground with scientific literature, since the unmutated subtype (UM-CLL) has been shown to exhibit a more aggressive phenotype compared to the mutated subtype (M-CLL) [68–72]. The activation of pathways such as Natural killer cell mediated cytotoxicity, B Cell Receptor Signaling, Focal Adhesion, and Negative Regulation of Cell-Cell Adhesion in UM-CLL reflect this aggressive nature. Specifically, the activation of immune-related pathways is associated with the intense inflammatory state, contributing to the aggressive responses of UM-CLL cells [68,69]. The increased cell mobility suggested by the activation of Focal Adhesion and Negative Regulation of Cell-Cell Adhesion pathways contribute to a more invasive and spreading phenotype [70,71].

We can also positively highlight that StellarPath found the B Cell Receptor (BCR) Signaling pathway activated. Particularly for the most aggressive unmutated subtype (UM-CLL), the B Cell Receptor (BCR) Signaling pathway plays a key role [73–75]. Differently from normal B cells, CLL's BCR signaling is characterized by low-level IgM expression, variable antigen response, and deregulation, contributing to disease development [73,75]. Elevated BCR signaling activity in CLL cells promotes survival and proliferation, and prognostic markers such as IgVH mutational status, ZAP-70, and CCL3 are associated with enhanced BCR signaling, suggesting a relationship with worse prognosis [74,75]. Furthermore, the PSN illustrated in Fig 5C thanks to StellarPath's ad-hoc function reveals more information about the individual patients. The patients UM.CLL4_2, UM.CLL4_4, UM.CLL4_13 and UM.CLL4_1 are the most important UM-CLL patients because are the most similar to the other UM-CLL patients and dissimilar to the M-CLL patients. The patient UM.CLL1_4, which also represents the similarities of UM.CLL4_6, UM.CLL4_71, UM.CLL4_83, UM.CLL1_4, UM.CLL1_6, UM.CLL1_7, UM.CLL2_1, UM.CLL3_1, UM.CLL3_4 (Table N in S1 Tables), looks like representing a subgroup of the UM class that regulates the pathway differently. Meanwhile, M-CLL patients are not cohesive, they are either isolated or close to UM-CLL patients. Finally, it is possible to observe that UM.CLL4_4, UM.CLL4_13 and UM.CLL4_1 are also important in the PSN representing the negative regulation of T cell activation (S2 File), suggesting that these patients are likely to be the most representative of the UM class.

## Computational resources

The patient similarity network paradigm brings advantages in the feature selection, the classification, and the overall interpretability of StellarPath. However, these benefits come at the cost of software scalability, a challenge previously highlighted by Pai et al. [1]. A single PSN in StellarPath represents a complete graph designed for a specific pathway. Consequently, increasing the number of patients or pathways requires greater memory RAM and extended running time for both storing and handling these networks. We measured the computational resources used by the classifiers (Fig 6) on a system with the following configuration: Intel i9-7900X 3.30GHz with 20 cores CPU, 120 Gigabytes of RAM, and Ubuntu 20.04 LTS.

StellarPath requires less running time and memory RAM than netDx (Table O in S1 Tables). This difference in resource usage between the two methods can be attributed to their unique implementations. StellarPath is implemented in R, naturally runs with multithreading, analyses the training data to minimize the number of PSNs, and integrates a Python module for the graph convolutional network's construction. On the other hand, netDx, designed in R and Java, does not analyze the training data to limit the PSNs. It opts to temporarily save networks to disk and only implements a sparsification approach on the PSNs to reduce their edge numbers.

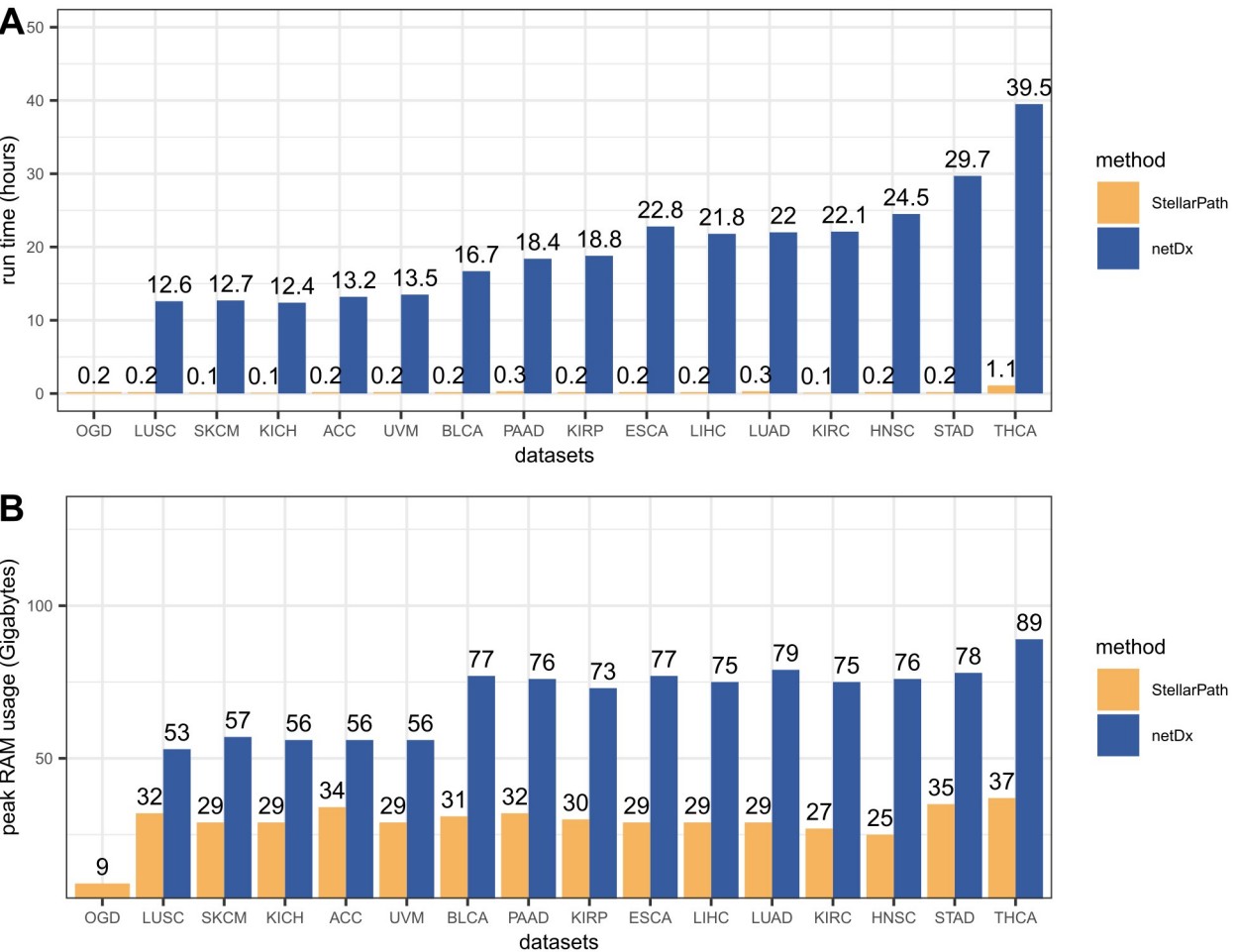

**Fig 6. Computational performances.** The bar plots show the computational resources required by StellarPath and netDx to classify the datasets. Running time is measured in hours and the memory RAM is measured in the maximum amount of Gigabytes that the software used. The X-axis indicates the datasets. The Y-axis indicates the measurement (Table O in S1 Tables).

## Discussion

We propose StellarPath, a deep learning patient classifier based on the pathway space and the patient similarity network paradigm that focuses on feature selection and interpretability. It analyses multi-omics datasets, performs a feature selection to identify markers, integrates the different omics in a hierarchical-vertical integration flow that exploits the interpretable pathway space, determines the similarity between patients in terms of their pathway activities, detects class-signature patient similarity networks and tests the selected molecules/pathways/PSNs to classify unknown patients. The method automatizes the biologist's task of classifying phenotypic conditions with pathways, inspects properties of molecules and regulatory networks that the human counterpart usually does not consider, and produces new information to further analyze the samples.

StellarPath gets further advantages from specific choices of software design that have not been previously taken into consideration with the PSN paradigm.

In the context of patient classification based on PSNs, StellarPath is the first method to adopt a hierarchical-vertical integration flow, to analyze omics data from normalization

through to integration in pathway-specific PSNs, and to adopt a novel similarity measure. This approach ensures software idempotence, reproducibility, and independence from user-specific settings. The results produced by StellarPath, including significant molecules, pathways, and PSNs, are more likely to align with findings from the literature. Additionally, this approach leads StellarPath to focus on a select set of likely-to-be-significant PSNs, which substantially reduces the software's computational requirements. In contrast, netDx, which neither processes the input data nor employs a specific similarity measure for each data type, cannot model omics' relationships as PSNs and can yield results that vary widely in meanings and values based on user choices regarding input data, parameters, and input functions.

StellarPath categorizes enriched pathways into four categories, compared to the standard two (up and down deregulation of the case class compared to the control one) used by traditional pathway analysis tools. The latter methods, like GSEA [19], calculate an enrichment score to discern up or down deregulated pathways between the two classes. This strategy assumes that the condition of the case patients is driving the deregulation of the enriched pathway, the differences between the two classes, and that the human expert has to interpret the score. The score by itself does not attribute the deregulation to a specific class and becomes extremely challenging to interpret when neither of the two classes includes real healthy samples (e.g. early versus late cancer stage, low versus high disease survival). For instance, a positive enrichment score can indicate both that the pathway is upregulated in the case patients and downregulated in the controls. StellarPath overcomes this limitation of traditional enrichment tools. It operates under the premise that the class with more stable molecules is the one primarily driving (by stress, disease, or phenotypic condition) the pathway's deregulation. This premise has been developed upon the following results. Genes with a low expression variance within a patient's class are significantly more likely to govern its phenotype [76–79] and to be core, connected, and regulated members of a functional pathway compared to genes with higher variance [31]. Genes that are different in terms of expression variance between two classes are not random and enrich pathways that correlate with phenotypic-specific mechanisms [31,80,81].

StellarPath employs a tailored similarity measure to compare patients' profiles in biological omics. This measure quantifies both how and how much two profiles regulate a specific pathway. As a result, the PSN generated by StellarPath mirrors how the patients regulate a significantly enriched pathway. This measure offers two advantages over traditional similarity measures such as the Euclidean distance, Pearson Correlation, and Cosine similarity (Section 1.6 in S1 Text). Firstly, it is easier to understand. The formula explicitly evaluates meaningful and interpretable criteria between molecular profiles. Secondly, it increases the likelihood of creating a patient similarity network that accurately captures the overall molecular differences between the two classes in a pathway. Furthermore, it also overcomes specific Pearson Correlation's limits such as being unable to consider the intensity and the disparity between the molecule's values describing two patients (Section 1.7 in S1 Text). StellarPath's similarity measure is not sensitive to outliers, accounts for the scale of the two profiles in comparison and considers both the direction and intensity of the patient's values in the same molecules.

Once StellarPath has created the PSNs, it analyses how much each patient is central to the similarities in its class. The StellarPath centrality measure is tailored for PSNs, which are unique in that they contain two distinct types of nodes (patients) and two types of relationships. This centrality measure captures the significance of a patient in relation to the two classes in comparison, operating under the assumption that a patient's importance is heightened when they are similar to patients within their own class and dissimilar to patients in the other class. Traditional measures, ranging from Degree Centrality's simple count of edges connected to a node, to Eigenvector Centrality's consideration of connections to high-scoring nodes, to

Subgraph Centrality's calculation of the number of closed walks for each node, provide valuable insights into many network contexts. However, they do not differentiate between the two types of nodes and relationships in the PSNs.

StellarPath employs a unique strategy by training a GCN on each pathway-specific PSN to classify unknown patients. It ensures consistent accurate prediction, reduces the chances of overfitting, enhances the reliability of the final classification, and improves the flexibility of the overall method. In fact, this strategy makes StellarPath flexible both technically and biologically. From a technical perspective, StellarPath can utilize either all or a subset of the trained GCNs, depending on the available data describing the unknown patients. If the unknown patients have the same types of omics data as the training patients (e.g., gene expression and miRNA), StellarPath can use all the trained GCNs. However, if the unknown patients have fewer types of omics data (e.g., only gene expression), StellarPath can still make predictions by utilizing only the applicable subset of GCNs. From a biological perspective, it allows StellarPath to embrace the fact that patients sharing a common clinical condition or phenotype, do not necessarily have identical genetic and epigenetics markers and do not necessarily show the same changes with respect to a control condition. As a consequence, an unknown patient can be predicted correctly without having to be necessarily similar to all the members of its class in all the pathways.

In this context, StellarPath's classification process does more than just predict a patient's class. It also provides an implicit prediction of the patient's molecular state and pathways. StellarPath tests whether a deregulated pathway is significant not only because it exhibits molecular differences between classes, as identified by functional enrichment analysis, but also because it can effectively distinguish these classes in completely unseen patients. While netDx does not test whenever the pathways or the data exhibit molecular differences between classes, and traditional functional enrichment analysis tools, such as GSEA, do not test the predictive power. StellarPath is designed to rigorously test both criteria, offering a more robust and clinically relevant analysis.

## Supporting information

**S1 Text. Supplementary information.** Text file that provides extra information and details regarding specific aspects, operations and concepts of the designed methods. Plus, it includes the rationale behind specific operations and implementations.
(PDF)

**S1 Tables. Supplementary tables and data.** Excel file that provides data, results and statistics produced with the method.
(XLSX)

**S1 File. Supplementary heatmaps.** File of heatmaps. An heatmap is associated to a specific dataset classified by StellarPath. An heatmap represents the patients of a dataset at the columns, the predictive and enriched pathways at the rows, an entry contains the centrality score of a column patient measured in pathway-specific (row) patient similarity network.
(PDF)

**S2 File. Supplementary PSNs.** File of patient similarity networks. Each PSN is the representation of the patient similarities related to a specific pathway which has been found enriched and predictive in classifying the CLL patients.
(PDF)

## Acknowledgments

We acknowledge Samuele Cancelleri for their constructive comments about this paper and work. Thanks to the Accelerating Research in Genomic Oncology-International Cancer Genome Consortium (ARGO-ICGC) for granting access to one of the CLL datasets used in the analysis (Study ID: EGAS00001000374 & Dataset ID: EGAD00001000258).

## Author Contributions

**Conceptualization:** Luca Giudice.

**Data curation:** Luca Giudice, Ahmed Mohamed.

**Formal analysis:** Luca Giudice.

**Funding acquisition:** Tarja Malm.

**Investigation:** Luca Giudice.

**Methodology:** Luca Giudice.

**Project administration:** Luca Giudice, Tarja Malm.

**Software:** Luca Giudice.

**Supervision:** Luca Giudice, Tarja Malm.

**Validation:** Luca Giudice.

**Writing – original draft:** Luca Giudice.

**Writing – review & editing:** Luca Giudice, Tarja Malm.

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
