## [Decision Letter · Decision Letter 0]

16 Jan 2024

Dear PhD Giudice,

Thank you very much for submitting your manuscript "StellarPath: hierarchical-vertical multi-omics classifier synergizes stable markers and interpretable similarity networks for patient learning" for consideration at PLOS Computational Biology.

As with all papers reviewed by the journal, your manuscript was reviewed by members of the editorial board and by several independent reviewers. In light of the reviews (below this email), we would like to invite the resubmission of a significantly-revised version that takes into account the reviewers' comments.

We cannot make any decision about publication until we have seen the revised manuscript and your response to the reviewers' comments. Your revised manuscript is also likely to be sent to reviewers for further evaluation.

Sincerely,

Andrey Rzhetsky

Academic Editor

PLOS Computational Biology

William Noble

Section Editor

PLOS Computational Biology

Reviewer's Responses to Questions

**Comments to the Authors:**

Reviewer #1: upload as an attachment

Reviewer #2: The paper presents an innovative method for patient classification and analysis, leveraging a hierarchical-vertical multi-omics classifier. The authors introduce StellarPath as a tailored approach for precision medicine, emphasizing its strong interpretability and potential impact on patient care. The manuscript provides a comprehensive overview of the method, its application in precision medicine, and the results of its validation. Overall, the paper presents a promising contribution to the field of patient classification and analysis.

The manuscript is well-structured and effectively communicates the novelty and potential impact of StellarPath. The authors have provided a clear rationale for the method, supported by relevant literature and a detailed description of the approach. The results and validation of StellarPath are presented in a coherent manner, contributing to the overall strength of the paper. However, there are some areas that require further attention and clarification.

Major Comments:

- Clarity and Structure: The introduction effectively sets the stage for the significance of patient classification and the limitations of existing methods. However, the introduction could benefit from a more explicit statement of the specific challenges that StellarPath aims to address.

- Interpretability and Clinical Relevance: The manuscript emphasizes the interpretability of StellarPath, which is a significant strength of the method. However, further discussion on how the interpretability of patient similarity networks translates to actionable insights for clinicians and researchers would enhance the clinical relevance of StellarPath.

Overall, the manuscript presents a promising method for patient classification and analysis, with potential implications for precision medicine. The authors have effectively communicated the novelty and significance of StellarPath, supported by a well-structured presentation of the method and its validation.

**Have the authors made all data and (if applicable) computational code underlying the findings in their manuscript fully available?**

Reviewer #1: Yes

Reviewer #2: Yes

PLOS authors have the option to publish the peer review history of their article (what does this mean?). If published, this will include your full peer review and any attached files.

Reviewer #1: No

Reviewer #2: No
---

## [Decision Letter · Decision Letter 1]

25 Mar 2024

Dear PhD Giudice,

We are pleased to inform you that your manuscript 'StellarPath: hierarchical-vertical multi-omics classifier synergizes stable markers and interpretable similarity networks for patient profiling' has been provisionally accepted for publication in PLOS Computational Biology.

Best regards,

Andrey Rzhetsky

Academic Editor

PLOS Computational Biology

Pedro Mendes

Section Editor

PLOS Computational Biology

Reviewer's Responses to Questions

**Comments to the Authors:**

Reviewer #1: All my comments and questions have been properly addressed. Good work.

**Have the authors made all data and (if applicable) computational code underlying the findings in their manuscript fully available?**

Reviewer #1: Yes

PLOS authors have the option to publish the peer review history of their article (what does this mean?). If published, this will include your full peer review and any attached files.

Reviewer #1: No

---

## [Editor Report · Acceptance letter]

4 Apr 2024

PCOMPBIOL-D-23-01808R1 

StellarPath: hierarchical-vertical multi-omics classifier synergizes stable markers and interpretable similarity networks for patient profiling

Dear Dr Giudice,

I am pleased to inform you that your manuscript has been formally accepted for publication in PLOS Computational Biology. Your manuscript is now with our production department and you will be notified of the publication date in due course.

With kind regards,

Anita Estes
